# Exploring Propolis as a Sustainable Bio-Preservative Agent to Control Foodborne Pathogens in Vacuum-Packed Cooked Ham

**DOI:** 10.3390/microorganisms12050914

**Published:** 2024-04-30

**Authors:** Eugenia Rendueles, Elba Mauriz, Javier Sanz-Gómez, Ana M. González-Paramás, Félix Adanero-Jorge, Camino García-Fernández

**Affiliations:** 1Institute of Food Science and Technology (ICTAL), La Serna 58, 24007 León, Spain; jjsang@unileon.es (J.S.-G.); fadaj@unileon.es (F.A.-J.); mc.garcia@unileon.es (C.G.-F.); 2ALINS—Food Nutrition and Safety Investigation Group, Universidad de León, 24007 León, Spain; 3GIP-USAL, Polyphenol Investigation Group, Universidad de Salamanca, 37007 Salamanca, Spain; paramas@usal.es

**Keywords:** cooked ham, propolis, food safety, natural additives, food control, ready-to-eat, meat

## Abstract

The search for natural food additives makes propolis an exciting alternative due to its known antimicrobial activity. This work aims to investigate propolis’ behavior as a nitrite substitute ingredient in cooked ham (a ready-to-eat product) when confronted with pathogenic microorganisms of food interest. The microbial evolution of *Listeria monocytogenes*, *Staphylococcus aureus*, *Bacillus cereus*, and *Clostridium sporogenes* inoculated at known doses was examined in different batches of cooked ham. The design of a challenge test according to their shelf life (45 days), pH values, and water activity allowed the determination of the mesophilic aerobic flora, psychotropic, and acid lactic bacteria viability. The test was completed with an organoleptic analysis of the samples, considering possible alterations in color and texture. The cooked ham formulation containing propolis instead of nitrites limited the potential growth (δ < 0.5 log_10_) of all the inoculated microorganisms until day 45, except for *L. monocytogenes*, which in turn exhibited a bacteriostatic effect between day 7 and 30 of the storage time. The sensory analysis revealed the consumer’s acceptance of cooked ham batches including propolis as a natural additive. These findings suggest the functionality of propolis as a promising alternative to artificial preservatives for ensuring food safety and reducing the proliferation risk of foodborne pathogens in ready-to-eat products.

## 1. Introduction

Food safety is one of the driving forces in recent research and development. Consumers claim natural foods and processed products that meet their organoleptic expectations. They also request ready-to-eat products that are compatible with their lifestyle, are nutritionally adequate, and do not pose health risks [1,2]. In this context, the food industry must comply not only with consumer trends but also with the legal requirements imposed by competent authorities, industrial benefits, and advances in food technology. On the other hand, there is a growing consumer demand for processed foods that provide increasingly varied sensory, usability, and shelf-life times, especially for products involving the manufacture of raw materials and hygienic and technological treatments. Considering these concerns, industry and science have worked together in search for novel additives that guarantee safety, preservation, and the development of sustainable products. Eventually, the advantages offered by these additives may provoke health disorders associated with allergic reactions, cardiovascular diseases, and cancer due to their abusive use and intake [3].

In this context, the upcoming research focuses on alternatives and solutions that avoid the undesirable effects of synthetic additives. These investigations must counteract the requirements of the food industry and the authority’s legislation to protect consumer health. Furthermore, the foods where artificial additives have been avoided or reduced (so-called Clean Label products) emerge in the consumers’ minds as being associated with safety, healthy food, and high-quality products [4,5].

The meat industry is one of the most affected by the current situation. The rise of artificial additives such as nitrites and nitrates is due to the perishable nature of raw meat together with the crescent demand for more stable products that conserve their organoleptic properties. The relevance of nitrites and nitrates as the most commonly used preservatives in the meat industry relies not only on the implementation of color, texture characteristics, odor, and flavor, but also, and most importantly, on controlling undesirable microorganisms, principally *Clostridium botulinum* [4,6,7]. As a result, the industry needs and authorities are forced to develop high-quality and safe meat products [8]. Thus, the European Food Safety Authority (EFSA) in the European Union [9] and the Food and Drug Administration (FDA) in the United States [10] have established restrictive intake limits. In this sense, scientific research identifies several natural substances as sustainable choices for the meat industry. Amongst natural preservatives, plant extracts have been the most traditionally studied due to their antimicrobial and antioxidant properties and because of their perception as being Generally Recognized as Safe (GRAS) [5,11]. Other natural conservatives considered potential substitutes for artificial additives include honeybee products.

From this perspective, propolis is a viscous resinous substance produced by *Apis melliphera* and other mellipones bees to defend the hive against parasites, bacteria, viruses, and other invaders. Bees make propolis from leaves, tree bark, and petals by mixing them with wax and then covering the cracks and holes [12]. Propolis, commonly known as bee glue, is rich in several bioactive compounds such as phenolic acids, flavonoids, and terpenes. The amount and variety of these substances are highly variated, depending on the botanic species around the hive and the season of harvesting, among others [13,14]. Antioxidant, anti-inflammatory, and antimicrobial capacities are the most appreciated properties of propolis. Due to the chemical structure and functional groups that conform to these bioactive molecules, propolis is a very aromatic substance with a high colorant capacity.

Although several studies have characterized propolis worldwide, assigning its composition to the bioactive profile, it is necessary to investigate propolis behavior profoundly when it is added to foods as a natural preservative [14]. Furthermore, propolis in the industry scale must be accompanied by studies that help understand propolis performance under real-production conditions [15,16]. In particular, the incorporation of propolis into the meat industry, specifically regarding cooked products, has not been comprehensively considered [17]. Some issues concerning the selection of propolis doses added to the food product without modifying their original color or taste are not sufficiently known [18]. Furthermore, the behavior of phenolic acids, flavonoids, and terpenes under high-temperature treatment needs to be studied thoroughly. The sensitivity patterns of foodborne pathogens against propolis depending on the matrix, process flow diagram, and growing conditions also deserve special attention [19]. This perspective offers an exciting standpoint for addressing the challenge of propolis becoming a natural alternative to artificial preservatives.

Therefore, the present study aimed to evaluate the effect of propolis extracts on the potential growth of *Listeria monocytogenes*, *Clostridium sporogenes*, *Bacillus cereus* [7,20], and *Staphylococcus aureus* in cooked ham formulations, as well as the consumer’s acceptance during defined shelf life.

## 2. Materials and Methods

### 2.1. Cooked Ham

#### 2.1.1. Ingredients and Procedure of Elaboration

The meat processing company established a recipe for cooked ham. A mixture of pork and fat at 82.64% was injected with brine prepared to 17%. The composition of this brine consisted of water, sodium chloride, potassium chloride, dextrose and stabilizers (triphosphates, natural thickener, potassium chloride, guar gum), antioxidants (sodium erythorbate), and preservatives (sodium acetate). The minced meat, brine, water, and ethanol extracts of propolis (EEP) were added to the mixer in the corresponding batches. The raw propolis concentration in the cooked ham was 0.81 g/kg of formulated cooked ham (27 mL of 30% (*w*:*v*) EEP). After 40 min of kneading, the mixture was left to rest for 8 h. Subsequently, it was stuffed into 5 kg pieces in triplicate for each formulation and adequately identified. It was cooled and packaged after cooking in three phases (55, 65, and 74 °C consecutively and ensuring 70 °C in the center of the piece).

#### 2.1.2. Nutrient Profile

The average compositional analysis of the product showed that per 100 g of edible fraction, there was 91.77 kcal/ 383.6 kJ, 18 g of protein, 1.3 g of carbohydrates, of which 1.3 g are sugars, 1.5 g of fats, of which 0.6 g are saturated fats, and 1.9 g of salt (NaCl). The sample batch to which nitrites were added was sodium nitrite (NaNO_2_), with a value of 13.92 ppm.

### 2.2. Propolis

Our previous research considered more than 30 propolis samples from the North of Spain, confronting all the EEP against the selected microorganisms based on their food industry relevance [21,22]. The analyses resulted in the selection of one of these 30 EEP sample according to their composition and biochemical profile, and most importantly to the sensitivity of the foodborne pathogens against EEP [22].

#### 2.2.1. Propolis Origin

The propolis extract considered to substitute nitrites in the formulation was from the North of León (Spain). The localization of the hives was informed by the beekeeper, who mixed the harvesting of each season of three close enclaves: 42.73642° N, 5.91060° W, 42.73896° N, 5.89276° W, and 42.75318° N, 5.88355° W. As described previously, the extract was prepared with 70% ethanol [21].

#### 2.2.2. Chemical Profile and Bioactive Compounds

The proximal composition in waxes, resins, impurities, and ashes, as well as minerals and humidity, were determined previously according to the methods recommended by Bankova et al. [23]. The equipment used for the different determinations and analyses were Soxtec System HT 1043 (Tecator, DK-3400, Hilleroed, Denmark) and Muffle Oven Mod. 10PR/300 Serial 88 (Hobersal, Barcelona, Spain). Furthermore, Polyphenols Total Content (PTC), Flavonoids Total Content (FTC), and Flavanones, Flavones, and Flavonols were also determined, as was explained in earlier works [24]. The extraction assays used ethanol from Panreac and methanol (Labkem, Barcelona, Spain), and a Spectrophotometer DU 7400 (Beckman, Brea, CA, USA) was used for all the assays. The preparation of the calibration graph involved the following products from different trading houses: Pinocembrin, Med Chem Express, and Galangin, Med Chem Express, both from Sweden and Quercetin, Sigma Aldrich (St. Louis, MO, USA). The antioxidant scavenging profile was also performed. Using different homologated Kits: ABTS Antioxidant Capacity Batch 10022604 and DPPH Antioxidant Capacity Batch 10072004. BQC Redox Technologies (Asturias, Spain) allowed us to test the EPP quickly and efficiently [25,26].

#### 2.2.3. HPLC-DAD-MS Quantification

Previosly, 100 mg of the sample was weighed and extracted with 30 mL of an EtOH:H_2_O mixture (70:30). The samples were then sonicated (Ultrasonic Cell Disruptor, MicrosonTM, Misonix, Atlanta, GA, USA) to be kept in an ultrasonic bath for 60 min. After completion, the samples were centrifuged for 10 min at 4000 rpm and the supernatant was carefully collected in a rotary evaporator balloon. The extraction was repeated twice. Finally, the three supernatants obtained for each of the samples were pooled and brought to dryness in a rotary evaporator at a temperature below 35 °C. The resulting residue was redissolved in 5 mL of MeOH:H_2_O (60:40) and kept for 2 min in the ultrasonic bath to favor its total solubility. Once the extract was obtained (and prior to its analysis) it was necessary to perform a filtration process (ClarinetTM, Hydrophilic PVDF 0.45 µm, Agela Technologies, Torrance, CA, USA) to then be injected into the chromatograph.

The analysis of phenolic compounds was performed by reversed-phase high-performance liquid chromatography, using on-line double detection by diode array spectrophotometry-mass spectrometry (HPLC-DAD-MS). The chromatographic equipment was a Hewlett-Packard 1200 (Agilent Technologies, Waldbronn, Germany) equipped with a binary pump and a diode array detector coupled to the HP Chem Station (rev. A.05.04). The separation was carried out on a Phenomenex Aqua^®^ C18 column (5 μm, 150 mm × 4.6 mm) thermostated at 35 °C, using 0.1% formic acid (eluent A) and acetonitrile (eluent B) as mobile phase. A flow rate of 0.5 mL/min was set, establishing the elution gradient. The injection volume was 15 μL and spectrophotometric detection was performed by selecting 280, 330, and 360 nm as preferred wavelengths.

Mass analysis was carried out using the mass spectrometer (Applied Biosystems, 3200 Q TRAP^®^ LC/MS/MS System, Waltham, Massachusetts, USA), operating in negative ionization mode at a temperature of 400 °C and recording spectra between *m*/*z* 100 and *m*/*z* 1000. Zero air was used as nebulizer gas (30 psi) and turbo gas (400 °C, 40 psi) for eluent drying, and nitrogen as curtain gas (20 psi) and medium collision gas [27].

The detection method employed was full scan at high sensitivity (Enhanced MS, EMS) with the following parameters: capillary voltage, −4500 V with the following potentials: declustering potential (DP) −50 V, entrance potential (EP) −6 V, and collision energy (CE) −10 V. Following this analysis, another analysis was carried out in Enhanced Product Ion (EPI) mode to obtain the characteristic fragmentation of the majority ion obtained in the first experiment. In this case the conditions used were: DP −50 V, EP −6 V, CE −25 V and collision energy spread (CES) 0 V. The identification of the phenolic compounds was carried out based on the retention time criteria observed in the chromatograms, the UV-visible spectra and the MS and MSn data obtained in the mass spectrometer, comparing the different data with those available in the literature. Peak quantification was compared with standards (p-coumaric acid, Galangin, and Chrysin) (standards’ mass spectra are included in Appendix A).

### 2.3. Inoculation of Strains

In the design of this study, the inoculation of pathogens of food interest from distinct groups was considered. Thus, based on the technological and legislative interest in cooked ham, the following microorganisms were selected: *Clostridium sporogenes* CECT 485 and 892; *C. sporogenes* is a non-toxingenic equivalent of some *C. botulinum* species, so we assumed the similar behavior of this microorganism in food to establish equivalence results [28,29]. A pool of five strains of *Bacillus cereus* was used to study the sensibility of this pathogen to the propolis sample (CECT 495, 8168, while 613, 635, and 553 were wild strains isolated from different meat products at our laboratory). The spore-form suspension was selected for the inoculation according to the optimum conditions to keep these spore-forming bacteria stable. The inoculation was performed in the cooked ham presentation since the preparation was fully manufactured in a food industry. Additionally, all the strains of *Staphylococcus aureus* (an enterotoxigenic food-borne pathogen) CECT 5190 and *Listeria monocytogenes* (CECT 10, 74, and 4032) were previously cultured in Mueller Hinton Broth (MHB, Biokar Diagnostics, Zac de Ther, Beauvais, France) at 37 °C for 24 h. After standardizing the content of viable cells, the concentrations (8 log_10_ CFU/mL after overnight culture) and inoculation volumes were established, generating a pool of strains. Portions of 25 g were meticulously cut from the whole cooked ham. Each portion was deeply cut and inoculated with 200 μL of the mixture of selected pathogenic microorganisms (inoculum weight/volume ratio was less than 0.01), ensuring the accuracy of our experiment (5 log_10_ CFU/g of each pathogen). At last, the samples were placed in a labeled sterile filter bag and vacuum-packed.

### 2.4. Experimental Design

For the design and development of the study, we contacted a meat industry that produced cooked ham. The disc diffusion method was used to evaluate the antibacterial activity of EEP. Petri dishes were prepared with Mueller Hinton Agar (MHA Biokar Diagnostics, Zac de Ther, Beauvais, France), and inoculated with bacterial culture in MHB before overnight incubation at 37 °C (2 × 10^8^ cfu/mL). Afterward, eight peripheral sterile Whatman filter paper discs with a symmetrical arrangement template and a central one were placed in each plate. Twenty µL of the EEPs were placed in the peripheral discs, using the same volume of 70% ethanol as a negative control. The growth of the inoculated strain was evaluated after incubation for 48 h at 37 °C (*L. monocytogenes* and *S. aureus*) and 30 °C (*B. cereus* and *C. sporogenes*), measuring the diameter of inhibition generated around the discs while using the diameter of inhibition of the central disc of each plate as a control. The diameter zones, including the diameter of the disc, were recorded. The minimum inhibitory concentrations (MICs) and minimum bacteriostatic concentrations (MBCs) were determined for the microorganisms selected against EEP based on previous studies [22,30]. From the EEP and based on the results of the composition analysis and determination of active compounds, further dilutions ensured a range concentration of 2500 µg of raw propolis/mL in the most concentrated dilution. V-bottom microtiter plates contained 50 µL of MHB in each cell. The MIC values determination, namely the minimum concentration that allows visible microbial growth, was achieved through known serial dilutions, using the last row of each plate as a growth control without EEP. The propolis’ concentrations inoculated ranged from 2500 µg/mL to 39.0625 µg/mL in the most diluted one. Finally, 50 of the bacterial inocula in MHB were included in each column and incubated at 37 °C for 48 h for *L. monocytogenes* and *S. aureus* (all covered with breathable rayon film), 30 °C for *B. cereus*, while *C. sporogenes* microtiter plates were incubated for 48 h under anaerobic conditions. In addition, complementary and parallel staining with Resazurin (Sigma Aldrich, St. Louis, MO, USA) at 0.01% showed a redox reaction, indicating the presence of viable cells in the dilution by a color change (naturally blue but turns purple). Results were obtained after 4 h of incubation. The concentration chosen to be incorporated in the meat product was established according to two essential parameters. The first issue was the antimicrobial property of the EEP against the different microorganisms (MIC and MBC), and the second issue was the cooked ham’s sensory characteristics, which wanted to be preserved [18]. The EEP was incorporated as an alternative to the nitrites that the industry contemplates in its commercial formulation. Thus, four formulations were considered: traditional recipe (A) with nitrites and with preservatives; (B) without nitrites and with preservatives; (C) without nitrites, with preservatives, with EPP; and (D) without nitrites, without preservatives, with EEP. Once elaborated, according to the industry’s recipe, each product was portioned under sterile conditions. A control group was established for each of them. On the other hand, the previously designed pool of pathogenic microorganisms was inoculated by applying it to the cooked ham. Three batches of each product and three samples of each batch per sampling point were prepared (according to the EFSA indications challenge test) [31]. The days of analysis were established considering the shelf life of the cooked ham indicated by the company (45 days). Thus, the following days were established: 0, 7, 15, 30, and 45. The storage temperature was 4 °C until day 30 and 8 °C until day 45 (Figure 1).

### 2.5. Microbiological Analyses

On the selected sampling days, 225 mL of Buffered peptone Water (BPW, Biokar Diagnostics, Zac de Ther, Beauvais, France) was added to each bag with *L. monocytogenes* and *S. aureus* pool and the control samples (without pathogens). Meanwhile, 225 mL of Reinforced Clostridia Medium (RCM, Biokar Diagnostics, Zac de Ther, Beauvais, France) is used for the ones with *Clostridium* and *Bacillus* and homogenized for 2 min (Masticator IUL, Barcelona, Spain) and decimally diluted in 9 mL of Bacteriological peptone 0.1% tubes (Oxoid LTD, Basingstoke, Hampshire, UK). Counts of *S. aureus* and *L. monocytogenes* were determined by spread plating various decimal dilutions on Baird Parker Agar (Biokar Diagnostics, Zac de Ther, Beauvais, France) and Listeria Brilliance Agar (Oxoid, Basingstoke, Hampshire, UK), respectively, and incubated at 37 °C for 48 h (Incubator Heraeus Instruments Function Line, Hanau, Germany). Similarly, *C. sporogenes* and *B. cereus* counts were conducted by properly plating various decimal dilutions on Gelose Sulfite de Fer (base TSC, Biokar Diagnostics, Zac de Ther, Beauvais, France) and Brilliance Bacillus cereus (BBc, Oxoid, Basingstoke, Hampshire, UK). TSC plates were incubated in anaerobic conditions by putting them in Oxoid Anaero Jar 2.5 L (Basingstoke, Hampshire, UK) with an AnaeroGen bag (Oxoid, Basingstoke, Hampshire, UK) at 30 °C for five days (Incubator Memmert INE-500, Schwabach, Germany). Conversely, BBc plates were incubated at 30 °C for 48 h (Incubator Memmert INE-500, Schwabach, Germany). The counts of these spore-forming bacteria included both populations (spores and vegetative cells). Some other microorganism groups were also considered in this assay because of the importance of technological or organoleptic role (Lactic Acid Bacteria, LAB), hygiene process information (*Enterobacteriaceae*), or even general population (Total Mesophilic and Psychrotrophic Aerobic Flora). These were counted using Plate Count Agar (PCA, Biokar Diagnostics, Zac de Ther, Beauvais, France), spreading deeply various decimal dilutions, then incubated at 30 °C for 48 h and 8 °C for 10 days, respectively. LAB enumeration on plates of Gelose MRS Agar (MRS, Biokar Diagnostics, Zac de Ther, Beauvais, France) incubated at 30 °C. Following the same scheme, *Enterobacteriaceae* in Violet Red Bile Glucose Agar (VRBGA, Biokar Diagnostics, Zac de Ther, Beauvais, France) at 37 °C for 24 h, with duplicates spread on every plate.

### 2.6. Physicochemical Analyses

The pH and water activity (Aw) were measured in all samples and batches throughout the storage, from the initial day to the last day of shelf life, which was established as an analysis point. pH values (pH Electrode, pH50 + DHS, XS Instruments, Carpi, Italy) were measured according to the Association of Analytical Communities (AOAC) guide and the Aw analyses (Water Activity meter LabMaster Aw, Novarasina, Neuheimstrasse, Lachen, CH, Switzerland) [32].

### 2.7. Sensory Analyses

A 10-member panel performed the sensory analysis using the homologated tasting room. The sheet was designed based on texture, color, and aroma parameters described as references in numerous tasting assays. The descriptors chosen were color, iridescence, consistency, own aroma, special aroma, hardness, juiciness, crumbliness, gumminess, cohesiveness, and fibrousness. The different formulations of cooked ham elaborated were considered. They should punctuate each descriptor with a score from 1 to 5. The participants knew the product well and were asked to develop another kind of test (the “Triangular Test (UNE-ISO 4120/2021)”) [33]. Three Triangular comparisons were conducted: (1) B vs. C, to evaluate the effect of the preservatives that were added; (2) B vs. D, to evaluate the effect of the EEP; (3) D vs. C, both batches with EEP but one of them without preservatives. These three tests aimed to discover if the panelists could find the influence of EEP on the sensory characteristics of cooked ham. Additionally, all panel members had the possibility of showing their verbal opinions. The samples were presented in individual dishes, with a homogeneous piece of 10.0 ± 0.1 g. The samples were at room temperature, and a white light regime was chosen. All the sensory analyses were performed in a testing room, with six places designed according to the UNE 87-004-79 Guide [34].

### 2.8. Statistical Analyses

Mean values ± standard deviation (SD) were used to represent continuous variables, while categorical variables were represented as absolute numbers and percentages. The normality of the data was assessed using the Kolmogorov-Smirnov test. When necessary, chi-square, Mann-Whitney U tests, and independent *t*-tests were employed to compare the variations between the organoleptic characteristics and exert control over the microorganisms in the different batches. The relationships between cooked ham formulations in terms of MIC and MBC values against foodborne pathogens were evaluated using one-way ANOVA and Kruskal-Wallis tests for normal and non-normal distributions, respectively. Data was analyzed using the SPSS for Windows version v.26 software program (IBM SPSS, Inc., Chicago, IL, USA). For every analysis, a *p*-value of less than 0.05 was designated as statistically significant.

## 3. Results

### 3.1. Propolis Chemical Profile and Bioactive Characteristics

#### 3.1.1. Physicochemical Characterization

A complete characterization of the EEP was performed at first. The bioactivity properties of this natural product are closely related to the resin fraction and waxes. The distribution of the proximal composition allowed the inclusion of the sample as the poplar propolis group. The values were expressed in percentage of raw propolis as follows: wax 19.80 ± 0.57, resin 60.5 2 ± 2.66, Ash 0.81 ± 0.04, impurities 0.06 ± 0.00, and moisture 12.87 ± 0.01. The mineral content was determined, too, in the raw propolis (RP). Potassium, calcium, and magnesium were the most abundant, with values of 2198.95 ± 6.93, 739.72 ± 8.51, and 221.55 ± 3.86 mg/kg, respectively [21].

Along with that, the deep study of the Resin fraction evidenced the content in polyphenols (CTP) 55.6 ± 1.2 g/100 g RP, flavonoids (CTF) 41.5 ± 0.43 g/100 g RP, flavones and flavonols (CFFT) 4.98 ± 0.25 g/100 g RP, and flavanones and dihydroflavanols (CFDT) 1.83 ± 1.64 g/100 g RP.

#### 3.1.2. Antioxidant Capacity

Different assays were performed to characterize the best possible antioxidant properties of the propolis. The values obtained using each method are shown in Table 1.

#### 3.1.3. Chemical Quantification of Phenolic Profile

The HPLC/MS analysis revealed the presence of 49 phenolic compounds in the EEP, including flavonol, phenolic acids, flavones, flavanones, and their derivates (Figure 2).

The flavonol content readout at 360 nm enabled the identification of 16 components (Table 2). Galangin was the majority compound, with a concentration of 19.30 ± 4.32 mg equivalents galanin/g raw propolis, followed by kaempferol and methyl quercetin (18, 19) (Figure 2a). The higher fraction of phenolic acids at λ = 330 nm (Table 3) was represented by the caffeic acid derivatives (peaks 1, 6, 28, 29, 33a, 35, 36, and 38). The caffeic acid benzyl ester compound showed a higher concentration, with 11.30 ± 1.11 mg equivalents p-coumaric acid/g raw propolis, while Caffeic Acid Prenyl Ester (CAPE) and its derivative showed 5.60 ± 0.28 and 4.40 ± 0.89 mg equivalents p-coumaric acid/g raw propolis, respectively. Furthermore, p-coumaric acid and derivatives (2, 24, 40, 44) represented 15% of the total hydroxycinnamic acids content. Ferulic and isoferulic acid (3, 4) were also found, although in lower concentrations (0.90 ± 0.08, 1.50 ± 0.10 mg equivalents p-coumaric acid/g raw propolis) (Figure 2b). At last, the flavones and flavanones group determined at λ 280 nm (Table 4) yielded remarkable concentrations of chrysin, pinocembrin, and pinobanksin-3-O-acetate, 13.60 ± 3.27, 16.40 ± 4.38, 13.60 ± 2.88 mg equivalents Chrysin/g raw propolis, respectively (Figure 2c).

### 3.2. Antibacterial Activity of EEP

#### 3.2.1. Inhibition Screening Test

The inhibition halos of each bacterial strain under assay were measured. All the pathogens showed sensibility against the EEP chosen. The halos ranged between 18 and 27 mm. The most sensitive to the EEP was *C. sporogenes*, while *L. monocytogenes* CECT 4032 and *B. cereus* CECT 635 presented the minimum halo (Table 5).

#### 3.2.2. MICs and MBCs Results for Each Foodborne Considered

The results for each microorganism for the minimum inhibitory and bactericidal concentrations are presented in Table 5 and Figure 3.

### 3.3. Challenge Testing

#### 3.3.1. Microbial Counts

Following the model designed for the challenge test (CT), the total counts of each pathogen and microorganism were performed along the five points of shelf-life storage. The inoculation rate of each food-borne bacteria was similar in all the batches and triplicates, ranging between 3.00 ± 0.10 and 4.00 ± 0.10 log_10_ CFU/g.

The behavior of *L. monocytogenes* in the cooked ham is modified according to the product’s formulation. The control batch (A) did not allow the *L. monocytogenes* proliferation throughout the storage time (δ = 0.39 log_10_). When nitrites are not included in the formulation and preservatives are the only additive (B), the potential growth (δ) raised until 4.58 log_10_ at the end of shelf life (45 days). The potential growth of *L. monocytogenes*, when batch D (EPP as the only preservative) is considered, followed a sawtooth pattern behavior: 2.36, 0.30, 0.00, and 0.71 log_10_ at the sampling time, respectively. Combining EEP and preservatives without nitrite (C) did not control *L. monocytogenes* growth from the 15 days of storage.

By contrast, the growth of *S. aureus* is negative until 15 days in refrigerated storage for the formulations considered. This is exemplified by batches A and D, with a 1.09 log and 1.19 log decrease, respectively. From day 30 to the end, microorganisms rose but were lower than the initial counts.

The spore-forming bacteria *B. cereus* did not exhibit an increase in the final shelf lifetime, except for batch C, which increased by 0.20 log_10_ CFU/g. The study of *C. sporogenes* revealed a drastic reduction in the counts in the four recipes, with values up to 3.00 log_10_ CFU/g. Both results refer to vegetative cells and spores since the storage conditions required temperatures below 8 °C to prevent germination. All results are shown in Figure 4.

The behavior of the indigene communities naturally in the cooked ham experimented with similar evolution during the 45 days of storage. *Enterobacteriaceae* did not grow from day 0 until day 45 in any sample. The rest of the groups considered counted were shown in Figure 5. The behavior of all the microorganism populations, either inoculated or indigenous strains, did not show statistically significant differences regarding the cooked ham formulation. On the other hand, the evolution of microbial communities along the storage time displayed differences statically significant regardless of the microorganism’s group: *L. monocytogenes* (*p* < 0.040); *S. aureus* (*p* < 0.001); *B. cereus* (*p* < 0.000); *C. sporogenes* (*p* < 0.008); mesophilic aerobic microorganisms (*p* < 0.001);
psychrotrophic microorganism (*p* < 0.001); and lactic acid bacteria (*p* < 0.001).

#### 3.3.2. pH and Water Activity

The pH values were like the four batches’ shelf life, showing non-statistically significant differences. The EEP batch displayed a drastic decrease in pH value from day 15 to the end of the storage time, exhibiting a minimum pH value of 5.37 ± 0.01. However, the other batches kept values close to six. When Aw was considered, the differences between batches were not statistically significant, independent of store time. As the days of storage elapsed, the evolution of Aw was similar in all samples. From day 15 to day 30, the Aw increased rapidly for the EEP + PRS and the NIT + PRS batches. The results obtained on the first day of the assay were 0.972 ± 0.002. The batch to which propolis had been added and nitrites and preservatives had been removed showed slightly higher values than the rest (in the range of 0.975 ± 0.001–0.976 ± 0.001) while the rest were between 0.969 ± 0.001–0.973 ± 0.001. Measurements did not show significant statistical differences in pH and Aw regarding the batch’s formulation. The results are shown in Figure 6.

### 3.4. Sensory Analyses Results

#### 3.4.1. Description of Cooked Ham Test

The panelist scored each codified batch for the parameters in the sheet. The consumer perception of discovering if the panelist could decline the batches with EEP was the principal objective of this test. All panelists accepted the attributes considered and did not report strange or undesirable organoleptic characteristics. Statistically significant differences were not observed except for color (*p* = 0.000), special flavor (*p* = 0.013), and crumbliness (*p* = 0.031). When considering the pairwise comparison of batches, the color showed no statistical differences between batches A and C (*p* = 0.436). Regarding the special aroma, the pairwise comparison between batches A–C and A–B showed statistical differences, with *p* = 0.007 and *p* = 0.020, respectively. As for the crumbliness, only pairwise C–D showed statically significant differences (*p* = 0.005) (Figure 7).

#### 3.4.2. Triangular Test

When this test was carried out, most panelists could find the sample different from the other two. Considering the first assay, 90% of the panelists found the correct one, as opposed to test two where only 70% got it right. Conversely, when the two batches with EEP were compared, all the panelists found a difference (Table 6). Statistically significant differences were not observed.

## 4. Discussion

This study investigates the possibility of applying propolis as a natural preservative in meat products. To date, most of the works focus on analyzing the behavior of food-interest microorganisms are from the in vitro perspective. Herein, we have investigated the feasibility of including propolis in different formulations of cooked ham as a sustainable alternative to nitrites or other commonly used artificial food additives.

The propolis selected in this study, among others from the region, is within the standard values criteria. In particular, the content of waxes, resins, humidity, and impurities followed the indications suggested by experts in honeybee products. Similarly, the bioactive compounds (polyphenols and flavonoids) quantified in the resinous fraction and the antioxidant capacity confirm the fundamental properties of propolis as a natural ingredient to replace traditional preservatives [35]. The natural characteristics of propolis have been previously studied, thus indicating the value of introducing constituents that appear in nature [13,36,37]. However, its use is limited to in vitro studies or raw meat products [1,38,39]. In this sense, this work provides an exciting pattern to incorporate propolis not only in non-heat-treated foodstuffs but also in ready-to-eat products such as cooked ham.

The quantification of the bioactive compounds by the HPLC method show that the quality of propolis added to the cooked ham formulation was good according to its phenolic profile. The content in flavonols (quercetin, galangin, and kaempferol), phenolic acids (p-coumaric acids and esters of caffeic and ferulic acids), and flavones and flavanones (chrysin and pinobanksin), are by previous works describing propolis from the exact geographical origin [5,24,40,41,42]. These compounds have demonstrated their bioactivity through a broad antimicrobial activity spectrum. For instance, the antibacterial activity of quercetin, p-coumaric acid, caffeic acid, and their derivatives occurred against bacterial species such as *S. aureus*, *L. monocytogenes*, and *B. cereus* [2,5]. Most flavonoids and phenolic acids identified in the selected propolis exhibited biological activity against Gram-positive and Gram-negative food-borne pathogens, as displayed by the low minimum inhibition concentrations [35]. Some flavonoids have even been considered for their effect on *C. botulinum* toxin production [14,43].

The observed antioxidant effect was additionally strengthened by the in vitro bactericidal capacity. First, the antimicrobial effect of propolis depends on several factors, including microorganisms’ type and cell concentration, in addition to the flavonoid content and botanical origin, as concluded in [14,38]. Our results evidence the high sensitivity towards propolis, as shown by the vast inhibition halos and the low MIC and MBC values obtained in the inoculated strains. These findings are in agreement with other works that reflect similar conclusions concerning the bacterial behavior of *L. monocytogenes* and *S. aureus* [44] and similarly with sporulated bacteria such as *B. cereus* [45] and *C. sporogenes* [46] at in vitro conditions [43].

In contrast, when the EEP was incorporated into the cooked ham formulation recipe, the microbial behavior was quite different than in vitro conditions. Propolis has been incorporated into meat products such as sausages [46], beef patties, salami [47], raw meat [39,48], and many dairy products such as ice cream, milk, and cheese [4,49,50,51]. However, when referring to foods subjected to heat treatment, current works only considered plant powder extracts as an alternative [14,39,43,52]. In this context, when considering the microbial behavior in cooked ham formulations, our results indicated that EEP can control the growth of sporulating bacteria (*C. sporogenes* and *B. cereus*) and *S. aureus* at low concentrations. The antimicrobial capacity of the EPP in cooked ham for *S. aureus*, *B. cereus*, and *C. sporogenes* showed a potential growth (δ) below 0.5 log_10_, thus indicating that the incorporation of EEP in the absence of nitrites avoids the microorganism’s development [4]. In the case of *B. cereus*, the growth control was even more effective in the batch that contained only propolis in the cooked ham formulation.

Nevertheless, *L. monocytogenes* exhibited an unexpected behavior with higher potential growth values (δ > 0.5 log_10_) in all the formulations except for the nitrites batch when considering their entire shelf lifetime (45 days). In this sense, Regulation (EC) No 2073/2005 [53] established that food can support the growth of *L. monocytogenes* when δ is higher than the limit of 0.5 log_10_. These results suggest that neither other preservatives nor propolis might control *Listeria*’s growth in the event of post-heat treatment contamination during its refrigerated storage. This discrepancy with the results of the MIC values for *L. monocytogenes* is probably due to the interference of the bioactive compounds with other food components and the effect of the processing (heat treatment) [43]. The same behavior for *L. monocytogenes* has been reported in another study investigating the bio-preservative properties of plant extracts in meat products after heat treatment [44,54]. Despite these considerations, it is essential to remark that this behavior did not occur in the batch containing only propolis (D). Although this batch experienced a drastic proliferation of *L. monocytogenes* in the first week of storage, the bactericidal effect of propolis controlled the growth within days 7 and day 30 of storage. Therefore, higher propolis concentrations or lower inoculating ranges will probably limit the rapid increment observed from the inoculation to day 7 of storage.

Finally, the evolution of indigenous microorganisms (mesophilic, psychrotrophic, and lactic acid bacteria) did not show differences between the batch formulations. Regarding mesophilic aerobic and psychrotrophic bacteria, other studies described similar growth trends throughout the storage time using comparable propolis concentrations [45,46].

Attending to the physicochemical characteristics of meat products, pH measurements, and activity water are critical to understanding the microorganism’s behavior as they determine the suitable environment for their growth. Despite not observing statistical differences between batches regarding pH values, formulation D, containing only propolis, experienced a soft decrease on day 30 that turned into a considerable drop at the end of the storage time. Although this event has not been sufficiently explored in previous studies, a similar pH decrease has been described by the effect of propolis and other plant extracts in Italian salami and chicken meat [6,44,55]. Therefore, when preservatives were maintained in batch C (propolis and preservatives), pH values remained stable due to the acidification control provided by the preservatives in the sample. Like the pH effect, the differences between batches regarding the Aw were not statistically significant. The same behavior was associated with the presence of preservatives in the formulations. Aw followed the same pattern in A and C batches when including either nitrite in the former one or preservatives and propolis in the latter. These batches experienced a remarkable increase from day 15 to day 30, when the storage temperature changed from 4° to 8 °C, followed by a drastic decrease until day 45 of storage. Conversely, batches B and D maintained stable values throughout the shelf life. However, these findings have no practical consequences for the meat industry since the amount of liquid phase added in the EEP (<0.01%) was meager, and the variations of Aw were within the expected ranges. Furthermore, there is no evidence of propolis action concerning Aw [4,11].

In addition to these considerations, the meat industry devotes special attention to the sensory characteristics bestowed on cooked ham by propolis. Propolis is, by nature, a very aromatic substance with a powerful coloring capacity [18,45,47]. For this reason, many previous studies have determined that propolis must not modify the food organoleptic characteristics when incorporated into food products. In this sense, although some tasters could discriminate between batches, only three descriptors presented statistical differences. First, tasters reported differences in the special aroma, although none found it inappropriate or unpleasant. Moreover, the aroma of cooked ham was always detected and not masked or hidden by the propolis. Concerning the color, the panelist did not find differences between the control batch and the batch of propolis and preservatives, thus suggesting the consumer’s acceptance of the product color (this observation is based on previous reports [16]). As for the crumbliness, there were only statistically significant differences between the batches containing propolis (D) and propolis and preservatives (C). This perception must be due to the absence of brine ingredients that confer texture properties to the cooked ham. At the same time, it confirms that integrating preservatives into the propolis recipe should be maintained to ensure palatability.

Despite these advances, this study presents several limitations. It is necessary to know the bacterial communities and their behavior in the cooked ham matrix in depth. Furthermore, the activity of the propolis extract needs to be evaluated in post-heat conditions since the bio-functional components (polyphenols and flavonoids) may suffer denaturation due to the elevated temperatures applied during the cooked treatment. Along with that, propolis might interact with other ingredients and additives included in food processing. Therefore, further studies will develop the inoculation in raw products instead of the foodstuffs. Overall, these results support the inclusion of propolis in cooked ham without interfering with the product’s taste, which is a promising use of ethanolic propolis extract as an alternative to traditional formulations.

## 5. Conclusions

Concern about the unfavorable effects of some artificial food additives on human health has triggered the search for other ingredients with similar functionalities. In this context, propolis represents an attractive alternative to substitute nitrites totally or partially as the traditional preservative in the meat industry.

This study presents the potential use of propolis in ready-to-eat products while ensuring food safety and preserving their conventional recipe. In particular, the addition of propolis in cooked ham formulations provides a protective effect against foodborne pathogens without altering their organoleptic features.

These results demonstrate the conservation capacity of propolis to limit the hazards of uncontrolled bacterial growth, except for *L. monocytogenes*, during meat product storage. Furthermore, the antioxidant properties confer propolis an additional benefit compared to the potential harmfulness of common food additives to consumers’ health.

This trend also reveals a future avenue of research by integrating preservatives of natural origin into sustainable food supplies. Therefore, the reformulation of meat products may not only reduce the risk associated with their contamination after manufacturing but also reinforce the consolidation of clean-label products as a realistic market choice.

## Figures and Tables

**Figure 1 microorganisms-12-00914-f001:**
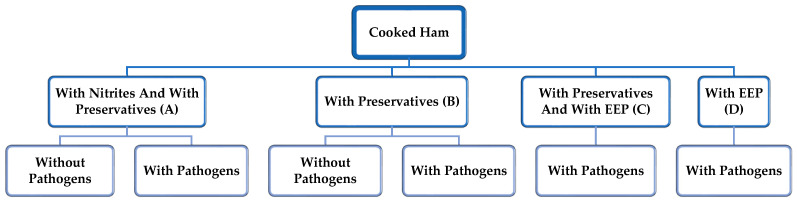
Experimental study design with six batches, according to formulation recipe and pathogens’ addition or not.

**Figure 2 microorganisms-12-00914-f002:**
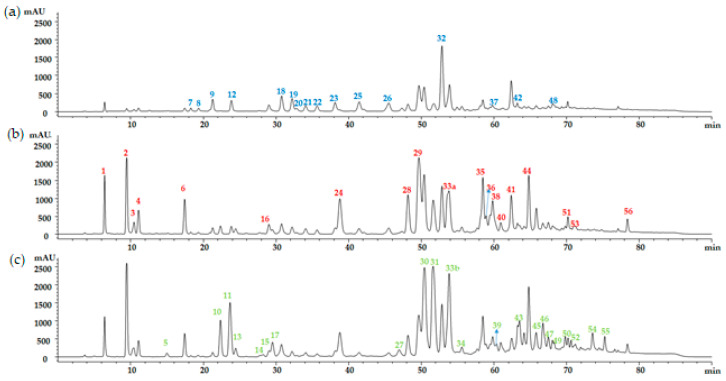
Chromatograph of ethanolic propolis extract monitoring for phenolic compounds: (**a**) 360 nm, (**b**) 330 nm, and (**c**) 280 nm.

**Figure 3 microorganisms-12-00914-f003:**
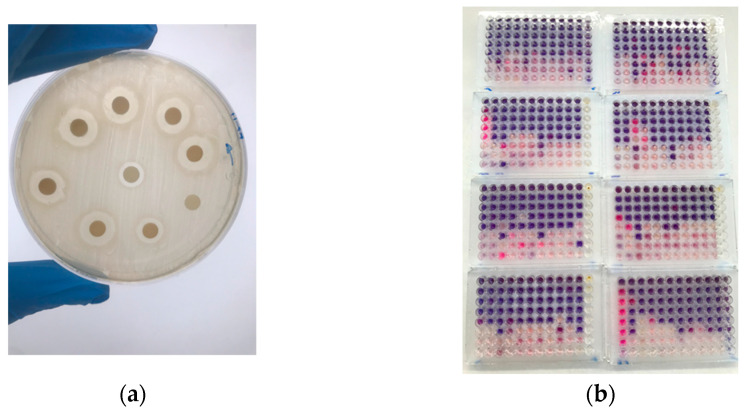
Representation of EEP inhibition capacity chosen against bacteria targets of assay. (**a**) Disc diffusion agar assay results *L. monocytogenes*; (**b**) microtiter plates and MIC and MBC tests (*B. cereus*, *C. sporogenes*, *L. monocytogenes*, and *S. aureus*).

**Figure 4 microorganisms-12-00914-f004:**
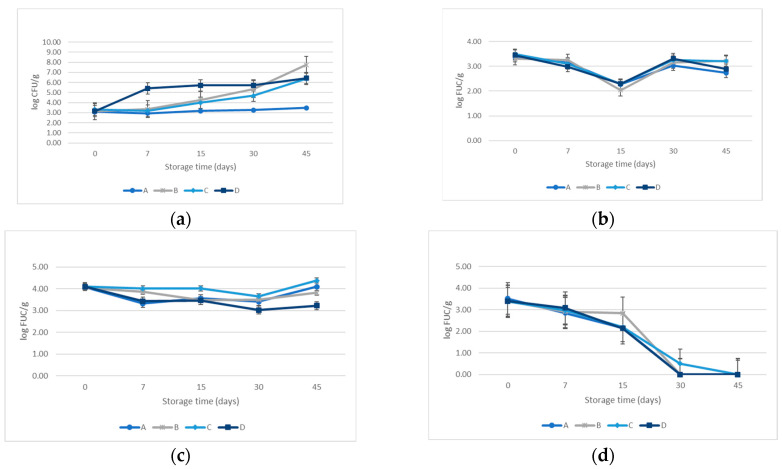
Evolution growth curves of inoculated bacteria. (**a**) *L. monocytogenes*; (**b**) *S. aureus*; (**c**) *B. cereus,* and (**d**) *C. sporogenes*.

**Figure 5 microorganisms-12-00914-f005:**
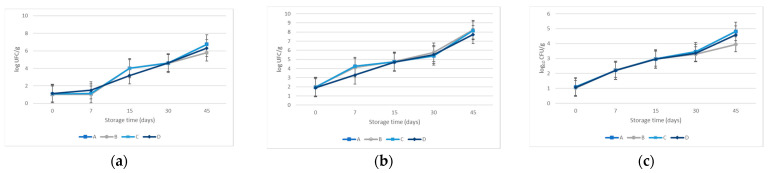
Evolution of indigenous microorganisms along storage time: (**a**) mesophilic aerobic microorganisms; (**b**) psychrotrophic microorganisms; (**c**). lactic acid bacteria.

**Figure 6 microorganisms-12-00914-f006:**
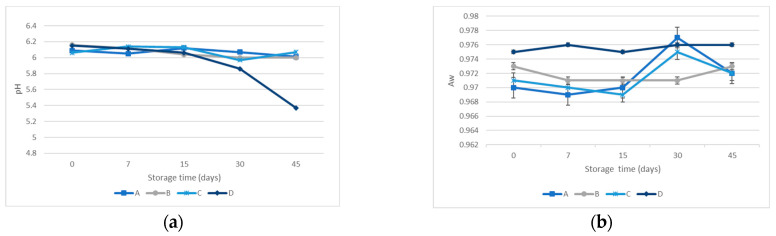
(**a**) Evolution of pH measurement along storage days. (**b**) Evolution of activity water measurement along storage days.

**Figure 7 microorganisms-12-00914-f007:**
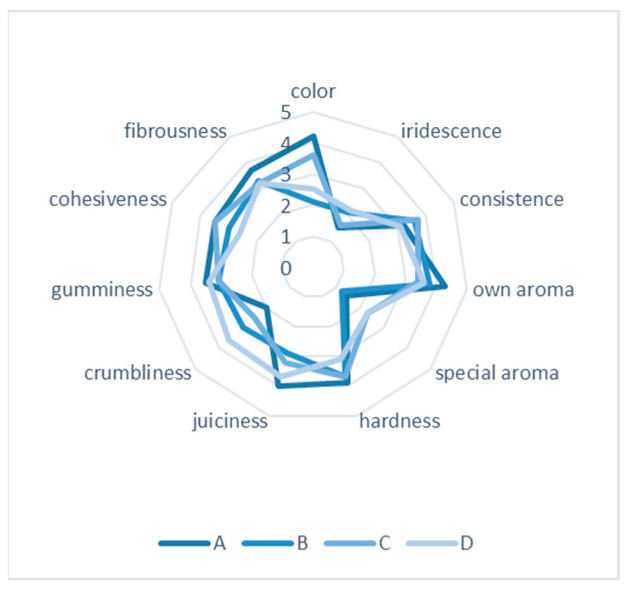
Sensory evaluation result of cooked ham samples. (A) Batch with nitrites and preservatives, (B) batch without nitrites and with preservatives, (C) batch with preservatives and with EEP, and (D) batch with EEP.

**Table 1 microorganisms-12-00914-t001:** Different assays determined antioxidant profile of propolis.

ABTS	DPPH	CEAC	FRAP
1653.46 ± 0.07	1115.38 ± 4.11	3800.44 ± 0.05	4188 ± 0.03
µM Trolox	µM Trolox	µM Vit C	µM Eq Galic acid ^1^

^1^ Results are related to raw propolis sample. ABTS: 2,2′-azino-bis (3-ethylbenzothiazoline-6-sulfonic acid); DPPH: 2,2-diphenyl-1-picrylhydrazyl; CEAC: Vitamin C Equivalents Antioxidant Capacity; FRAP: Ferric Reducing Antioxidant Power.

**Table 2 microorganisms-12-00914-t002:** Total flavonol content expressed as mg equivalents galangin/g raw propolis (λ = 360 nm).

Peak	Component	RT (min.)	[M - H] (*m*/*z*)	MS^2^ (*m*/*z*)	Amount
7	Methylluteolin	19.4	299	284, 255, 227	0.90 ± 0.12
8	Dimethylquercetin	21.3	329	315, 299, 285	0.80 ± 0.15
9	Quercetin	22.4	301	179, 151	4.10 ± 0.52
12	Methylquercetin	24.4	315	301, 271, 255	3.40 ± 0.34
16	Apigenin	29.5	269	225, 180, 149, 117	2.60 ± 0.34
18	Kaempferol	32.2	285	257, 229, 151	5.80 ± 0.78
19	Methylquercetin	32.8	315	301, 151	4.50 ± 0.66
21	Methylluteolin (Luteolin-Methyl-Ether)	34.1	299	284, 255, 227	2.10 ± 0.35
22	Methoxykaempferol 3-Methyl Ether	35.6	329	314, 299, 285	2.00 ± 0.39
25	Quercetin-7-Methyl-Ether	41.5	315	301, 193, 165, 121	3.70 ± 0.66
26	Quercetin-Dimethyl-Ether	45.5	329	315, 299, 271	3.50 ± 0.71
32	Galangin	52.8	269	227, 197	19.30 ± 4.32
37	Galangin-5-Methyl-Ether	59.5	283	268, 177, 133	1.50 ± 0.83
42	Luteolin 6-C-Pentoside (Arabinoside)	63.2	421	313, 299	1.00 ± 0.29
48	Galangin-5-Methyl-Ether	68.1	283	268, 239	<DL
	Total Flavonols Content				56.10 ± 10.02

RT: Retention Time; DL: Detection Limit.

**Table 3 microorganisms-12-00914-t003:** Total Phenolic Acids Content express as mg equivalents p-coumaric acid/g raw propolis (λ = 330 nm).

Peak	Component	RT (min.)	[M - H] (*m/z*)	MS^2^ (*m*/*z*)	Amount
1	Caffeic Acid	6.4	179	135	3.80 ± 0.19
2	p-Coumaric Acid	9.5	163	119	7.60 ± 0.58
3	Ferulic Acid	10.5	193	178, 149, 134	0.90 ± 0.08
4	Isoferulic Acid	11.1	193	178, 134	1.50 ± 0.10
6	3,4-Dimethyl-Caffeic Acid (DMCA)	18.3	207	163, 133	3.50 ± 0.23
24	Coumaric Acid Derivative	38.8	301	165, 135	5.70 ± 0.77
28	Caffeic Acid Prenyl Ester (CAPE)	48.2	247	179, 135	5.60 ± 0.28
29	Caffeic Acid Benzyl Ester	49.7	269	168, 161, 134	11.30 ± 1.11
33a	Caffeic Acid Phenylethyl Ester (CAPE)	53.8	283	179, 161, 135	6.00 ± 1.13
35	Caffeic Acid Cinnamyl Ester	58.5	295	178, 134	5.90 ± 1.23
36	Caffeic Acid Methyl Phenetyl Ester	59.0	297	179, 161, 135	<QL
38	CAPE Derivative	59.8	551	429, 283, 267, 255	4.40 ± 0.89
40	Coumaric Acid Derivative	61.0	267	163, 145, 119	1.30 ± 0.19
41	Methoxychrysin Derivative	62.4	301	283, 269, 253, 152	1.80 ± 0.36
44	P-Coumaric Cinnamyl Ester	64.8	279	235, 195, 118	3.40 ± 1.04
	Total Hidroxicinamics Acids Content				62.6 ± 7.82

RT: Retention time; QL: quantification limit.

**Table 4 microorganisms-12-00914-t004:** Total flavones and flavanones content expressed as mg equivalents chrysin/g raw propolis (λ = 280 nm).

Peak	Component	RT (min.)	[M - H] (*m*/*z*)	MS^2^ (*m*/*z*)	Amount
5	Genistein glucoside	17.4	431	268, 239	<LD
10	Methylpinobanksin	23.7	285	267, 253	5.40 ± 0.31
11	Sakuranetin	23.8	285	267, 251	9.00 ± 0.60
13	Methylapigenin (Ej. Hispidulin)	27.8	299	270, 255	2.50 ± 0.42
14	Methylchrysin	28.2	267	252, 224, 180	<LD
15	Pinobanksin Derivative	29.1	271	177, 151, 119	<LD
17	Pinobanksin	30.8	271	253, 197	3.70 ± 0.47
23	Methoxy-Chrysin	38.1	283	268, 239, 211	<LD
27	Pinobanksin-5-Methyl-Ether	47.0	287	193, 181, 166	0.90 ± 0.33
30	Chrysin	50.5	253	209, 167	13.60 ± 3.27
31	Pinocembrin	51.7	257	255, 213, 151	16.40 ± 4.38
33b	Pinobanksin-3-O-Acetate	53.8	313	271, 253	13.60 ± 2.88
34	Methoxy-Chrysin	55.6	283	268, 239	0.80 ± 0.16
39	Pinobanksin-5-Methyl-Ether-3-O-Acetate	60.9	327	271, 253	<LD
43	Pinobanksin	63.5	271	253, 165, 152	3.70 ± 0.97
46	Pinocembrin Derivative	66.7	363	269, 257	
49	Naringenin	68.3	521, 271	283, 269	<LD
50	Pinobanksin-3-O-Pentanoate or 2-Methylbutyrate	69.8	355	271, 255	<LD
	Total Flavones and Flavanones				73.30 ± 13.70

RT: Retention time; LD: detection limit.

**Table 5 microorganisms-12-00914-t005:** Antibacterial activity of ethanol extract of propolis (EEP).

Microorganisms	Strains	Halo (mm)	MIC (µg/mL)	MBC (µg/mL)
*Listeria monocytogenes*	10	17	312.50	625.00
	74	17	156.25	312.25
	4032	18	156.25	312.25
*Staphylococcus aureus*	5190	25	625.00	625.00
*Clostridium sporogenes*	485	27	156.25	312.50
	892	26	625.00	625.00
*Bacillus cereus*	495	22	312.50	625.00
	553	21	312.50	1250.00
	613	20	312.50	625.00
	635	18	312.50	625.00
	8168	25	312.50	625.00

MIC—Minimum inhibitory concentration; MBC—minimum bactericidal concentration.

**Table 6 microorganisms-12-00914-t006:** Results obtained after triangular test sensory evaluation of three dishes.

Participants										
Triangular Test	1	2	3	4	5	6	7	8	9	10
Dish 1	☑	☑	☑	☑	☑	☑	☑	⊠	☑	☑
Dish 2	☑	☑	☑	☑	☑	☑	⊠	⊠	⊠	☑
Dish 3	☑	☑	☑	☑	☑	☑	☑	☑	☑	☑

☑ Correct and ⊠ incorrect answer.

## Data Availability

Data are contained within the article (and Appendix A).

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
