# Peer review of "Exploring Propolis as a Sustainable Bio-Preservative Agent to Control Foodborne Pathogens in Vacuum-Packed Cooked Ham"

_microorganisms, 2024, doi:10.3390/microorganisms12050914_

Round 1

Reviewer 1 Report

Comments and Suggestions for Authors

Comments to the Author

I have completed my evaluation of the manuscript titled " Exploring propolis as a sustainable bio-preservative agent to control foodborne pathogens in cooked ham " by Eugenia Rendueles et al. Here are some questions and suggestions I have for this paper.

1.    Please describe the abbreviations that appear for the first time in the manuscript (full English name), e.g., EPP, BPW. Additionally, abbreviations in the tables in section 3.1.2 can be introduced in the results section.

2.    I suggest extracting the mass spectra peaks of key compounds or annotating them in the total ion chromatogram. Supplement the standard substance spectra.

3.    Please optimize the quality of the figures in the manuscript.

4.    The probability value "p" in line 391-393 should be italicized.

5.    Please add figures or tables to section "3.4.2". Additionally, it is recommended to provide evidence of sensory evaluation (such as surveys or other materials) and upload them as attachments.

Comments on the Quality of English Language

Minor editing of English language required.

Reviewer 2 Report

Comments and Suggestions for Authors

1.      Lines 97-103 “The materials and equipment used in this experiment” please omit this paragraph and write the company and country of each one at its mention in the text”

2.      Line 105: medium instead of “mediums” and write the company and country of each one at its mention in the text.

3.      Also, the equipment used for the different determinations and analyses should be mentioned one by one according to methods.

4.      Line 149 “EEP” & L 230 “EFSA” & 262 “AOAC” please write in full at the first mention then write the abbreviation.

5.      Line 200: replace “spice” with “species.”

6.      Lines  203, 206: and “not italic”

7.      Line 206: replace “Cultured in” with “cultured on”

8.      L 216” minimum inhibitory and bacteriostatic concentrations “MIC and MBC”” replace it with “Minimum inhibitory concentrations (MICs) and minimum bactericidal concentrations (MBCs)”. More details should be mentioned (media, concentration range).

9.      L 240, 241: L. monocytogens, S. aureus

10. Legend of figure 3 should be more detailed with indication of M.O, different discs in the plate. Also, the same regarding MIC. One microtiter plate is sufficient to indicate MIC and MBC values.

11. L 361 “food-bone”

12. L 391-393& 419-424: “p” should be italicized

Comments on the Quality of English Language

Minor editing of English language required

Reviewer 3 Report

Comments and Suggestions for Authors

the title accurately reflects the content of the article
The abstract summarizes the most important points and contributions and the objectives are clearly stated.
The paper describes well the methods used, but does not include references or a more detailed description of the methods used that lead to Table 5 and Figure 3 a and b.

The figures and tables are well designed and the results are clearly visible.

The paper is a new contribution to field research and offers practical conclusions or recommendations for the future.

Line: 148:  the phrase “ After many tests and assays”, should be rephrased and include references. 

Line: 219 - the phrase “ according to two critical parameters”, should include which are these parameters, including a reference. 

Line: 469 - the phrase "as conclude in numerous studies” when  only presented two references,  should be rephrased. 

Reviewer 4 Report

Comments and Suggestions for Authors

Searching for alternatives to nitrites and nitrates in foods is an important objective; the authors propose the use of EEP for the control of some pathogens in sliced cooked ham; the manuscript is very interesting, but I think the authors need to clarify some aspects of the challenge

I suggest integrating the title  “Exploring propolis as a sustainable biopreservative agent for the control of foodborne pathogens in vacuum-packed cooked ham slices

Line 57 : principally Clostridium botulinum [4,6].As - space

Line 91: Therefore, the present study aimed to evaluate the effect of propolis extracts on the potential growth of Listeria monocytogenes, Clostridium sporogenes, Bacillus cereus, and Staphylococcus aureus in cooked ham ….

The authors should support the choice of these pathogens with some epidemiological data and references, particularly for Clostridium botulinum and B. cereus

Did the authors inoculate the spores of Clostridium sporogenes and B. cereus?

Line 134: The minced meat, brine, water, and EEP

EEP, the first time in extensor

Line 196: 2.4. Inoculation of strains

Have the pathogens of interest been searched for in the raw product?

Line 138: 70ºC

70 °C

Line 199: C. sporogenes is an apathogenic spice of C. botulinum, so we assumed the similar behavior  of this microorganism in food to establish equivalence results.

This sentence must be supported by references

2.5. Experimental design

Meat products and spices can frequently be contaminated with Bacillus cereus and Clostridium botulinum spores; nitrites are added to prevent spore germination; in the case of cooked ham the spores can resist heat treatment, but are inhibited by the presence of nitrites; for this reason I would have expected the authors to inoculate the spores of B. cereus and C. botulinum before cooking, but instead they inoculated them on the slices; I don't understand the challenge test of these pathogens in sliced cooked ham; testing for L. monocytogenes and S. aureus is appropriate; in fact these pathogens are inactivated by heat treatment, but often recontaminate the product after slicing; moreover, the sliced cooked ham is packaged in a modified atmosphere

216: The concentration chosen to be incorporated in the meat product was established according to two critical parameters.

 The authors must indicate the amount of EEP added/kg and the methods

Line 230: 0, 7, 15, 15, 30 and 45

15 it's a repetition

Line 245: counts were conducted by properly plating various decimal dilutions on TSC

No italic

Figure 1:  sliced cooked ham

Line 358: inoculation rate of each food bone

food borne

Line 369: the growth of S. aureus is negative until 15 days in freeze storage

Have the samples been frozen?

Line 373: B. cereus  Italic

Line 472: such as sausage- sausages

Line 519: from 4º to 8ºC, 4 ° to 8 °C

Line 559: These results demonstrate the conservation capacity of propolis to limit the hazards  of uncontrolled bacterial growth during meat product storage.

not for L. monocytogenes; the sentence should be reformulated
